# Genetic Characteristics and Long-Term Follow-Up of Slovenian Patients with RPGR Retinal Dystrophy

**DOI:** 10.3390/ijms24043840

**Published:** 2023-02-14

**Authors:** Vlasta Hadalin, Maša Buscarino, Jana Sajovic, Andrej Meglič, Martina Jarc-Vidmar, Marko Hawlina, Marija Volk, Ana Fakin

**Affiliations:** 1Eye Hospital, University Medical Centre Ljubljana, Grablovičeva 46, 1000 Ljubljana, Slovenia; 2Clinical Institute of Genomic Medicine, University Medical Centre Ljubljana, Šlajmerjeva 4, 1000 Ljubljana, Slovenia

**Keywords:** RPGR, rod-cone dystrophy, cone dystrophy

## Abstract

Genetic characteristics and a long-term clinical follow-up of 18 Slovenian retinitis pigmentosa GTPase regulator (RPGR) patients from 10 families with retinitis pigmentosa (RP) or cone/cone-rod dystrophy (COD/CORD) are reported. RP (eight families) was associated with two already known (p.(Ser407Ilefs*46) and p.(Glu746Argfs*23)) and five novel variants (c.1245+704_1415-2286del, p.(Glu660*), p.(Ala153Thr), c.1506+1G>T, and p.(Arg780Serfs*54)). COD (two families) was associated with p.(Ter1153Lysext*38). The median age of onset in males with RP (N = 9) was 6 years. At the first examination (median age of 32 years), the median best corrected visual acuity (BCVA) was 0.30 logMAR, and all patients had a hyperautofluorescent ring on fundus autofluorescence (FAF) encircling preserved photoreceptors. At the last follow-up (median age of 39 years), the median BCVA was 0.48 logMAR, and FAF showed ring constriction transitioning to patch in 2/9. Among females (N = 6; median age of 40 years), two had normal/near-normal FAF, one had unilateral RP (male pattern), and three had a radial and/or focal pattern of retinal degeneration. After a median of 4 years (4–21) of follow-up, 2/6 exhibited disease progression. The median age of onset in males with COD was 25 years. At first examination (median age of 35 years), the median BCVA was 1.00 logMAR, and all patients had a hyperautofluorescent FAF ring encircling foveal photoreceptor loss. At the last follow-up (median age of 42 years), the median BCVA was 1.30 logMAR, and FAF showed ring enlargement. The majority of the identified variants (75%; 6/8) had not been previously reported in other *RPGR* cohorts, which suggested the presence of distinct *RPGR* alleles in the Slovenian population.

## 1. Introduction

### 1.1. Molecular Genetics of the RPGR Gene

The retinitis pigmentosa GTPase regulator *(RPGR*) is a gene located in the Xp11.4 chromosomal region that spans 172 kbs and up to 22 exons depending on splicing [1,2]. The RPGR protein, which is expressed in different tissues of the human body such as those of the lung, kidney, testis, brain, and retina [3] is located in the transition zone of primary and motile cilia and at the centrosomes and centrioles in dividing cells [3,4,5]. Its exact role in the retina has not yet been fully elucidated; however, it has been suggested that it plays a critical role in ciliary genesis, maintenance, and functions such as protein trafficking and sorting [6,7]. RPGR’s function as a guanine exchange factor (GEF) for small guanosine triphosphatases (GTPases) may play an important role in cargo trafficking to the photoreceptors’ outer segments [6,8]. It has been shown that without the RPGR protein, mouse photoreceptor cells develop normal morphology and are able to carry out phototransduction and remain viable in the first few months of life. However, a loss in RPGR alters protein trafficking to the photoreceptors’ outer segment over time, and partial mislocalization of opsins apparently reduces the viability of photoreceptors [6,9,10,11]. Therefore, the ciliary function of RPGR does not appear to be central but instead facilitative and is required for the long-term maintenance of photoreceptors [9]. Due to alternative splicing, more than 20 different RPGR isoforms have been identified [3,12,13]. There are three major isoforms found in the retina: *RPGR*^ex1–19^, *RPGR*^ORF15^, and *RPGR*^skip14/15^. *RPGR*^ex1–19^ is derived from exons 1–19. *RPGR*^ORF15^ is derived from exons 1–14 shared with the *RPGR*^ex1–19^ isoform and the additional exon ORF15, which contains exon 15 and a part of intron 15 at the 3′ end [14]. The recently identified *RPGR*^skip14/15^ is generated via alternative splicing of exons 14 and 15, which leads to an in-frame deletion in RPGR transcripts [15]. Although the *RPGR*^ORF15^ isoform contains a lower number of exons as compared to the *RPGR*^ex1–19^ isoform, exon ORF15 is longer than the combined length of exons 16–19. This exon also has an unusual composition of purine-rich repeats that encompass 1.5 kb and encode a protein domain of 560 amino acids. This domain is followed by a short stretch of basic amino acids that are also called the RPGR-C2 domain (1071–1152 amino acids). The Glu-Gly domain of RPGR exhibits sequences similar to those of the polyglutamated regions of alfa-tubulin [10]. RPGR^ORF15^ glutamylation is regulated by the tubulin-tyrosine ligase-like 5 (TTLL5) enzyme, which has also been associated with retinal dystrophies [16].

### 1.2. Retinal Dystrophies Associated with the RPGR Gene

Pathogenic variants in *RPGR* result in different retinal disorders; these are most commonly retinitis pigmentosa (RP) (70–90%) and less frequently cone dystrophy (COD) (7%) or cone-rod dystrophy (CORD) (6–23%) [17,18,19]. Many studies have suggested that the existence of two contrasting disorders may depend on the location of the variant: variants in exons 1-14 and the proximal part of the ORF15 exon usually result in RP, while variants in the distal end of the ORF15 exon cause COD/CORD. The *ORF15* exon is considered a mutational hot spot because it encodes a highly repetitive domain and most of the disease-associated variants are truncating [20,21,22]. As described by De Silva and colleagues, there is a watershed zone of approximately 100 amino acids between the two regions where variants can result in either phenotype. 

RP is a group of genetically different diseases that together represent the most common inherited retinal dystrophy (affecting 1 in 3000 people) [23]. The main feature is an irreversible loss of photoreceptors (rods and secondary cones) that is caused by more than 70 different genes (Retinal Information Network (Retnet); https://sph.uth.edu/retnet, accessed on 15 November 2022) and that can be inherited in autosomal recessive, autosomal dominant, mitochondrial, or X-linked fashions [24]. The latter inheritance is found in 5–15% of RP cases [25] and is most often (>70%) associated with pathogenic variants in *RPGR* [17]. X-linked RP (XLRP), which includes RP caused by variants in the *RPGR* gene (RPGR-RP) (including a hotspot in the ORF15 exon), is usually more severe than other RP types [21]. It usually presents in early childhood with nyctalopia and peripheral visual field (VF) loss. The majority of patients progress to legal blindness, which is usually defined as either a visual acuity (VA) of logMAR ≥ 1.0 (≤20/200 Snellen equivalent) or a VF diameter ≤20° [26] in the fourth decade, although studies have reported varying timescales [18,27].

Although X-linked inheritance is rare in COD/CORD (only 1%) [28], the *RPGR* gene is responsible for 73% of COD/CORD [29]. The main initial symptoms are reduced VA and abnormal color vision, central scotoma, and photophobia. The symptoms are associated with variants in all isoforms, although there are conflicting reports regarding the association between the location of the variant within the *RPGR* gene and the disease severity (for further elaboration, see the Discussion section). Signs of rod dysfunction may occur with disease progression, and patients may also experience night blindness and peripheral VF loss [18,30]. In contrast to RP, the onset of symptoms occurs later (in the fourth decade) [18,31], but can progress to blindness relatively quickly (at the age of 40–50 years) [18,31]. In a study by Nassisi M and colleagues, the rate of the best corrected VA (BCVA) decline was assessed at about 7% per year, and most patients reached a BCVA ≥ 1 logMAR during the fifth decade of life [32].

Most RPGR patients have myopia; this includes those with RP and those with the COD/CORD phenotype [21]. A correlation between a high myopic refractive error and faster rates of VA loss in all phenotypes has been reported, but this remains debatable [33].

Due to X-linked inheritance, the disease predominantly affects males, but females can also be affected (often with significant asymmetry between the eyes). The possible reasons may be the pattern of random inactivation of the X chromosome, gene dosage (females have two copies), and other genetic and environmental factors that determine the phenotype expressed [34]. Female phenotype patterns are classified based on fundus autofluorescence (FAF) images as the following patterns: normal or near normal (mild retinopathy) fundus appearance (normal/near-normal pattern), spoke-shaped reflexes extending from the central macular area in a radial pattern (radial pattern), focal pigmentary retinopathy and/or patchy pigmentation with a radial reflex pattern (focal pattern), and male pattern retinitis pigmentosa (male pattern) [35]. Slit lamp fundus features are also used: grade 0 (no fundus abnormalities), grade 1 (a tapetal-like reflex without pigmentary changes in the retina), grade 2 (regional pigmentary changes; e.g., bone spicule-like pigmentation involving at least two quadrants and/or macular retinal pigment epithelium (RPE) alterations with or without a tapetal-like reflex), and grade 3 (at least three quadrants of pigmentary changes or RPE atrophy in the periphery) [36,37,38]. Phenotypes of tapetal reflex or focal pigmentary changes are described to be the most frequent [35]. As in males, in females an increasing age was associated with a lower BCVA and constricted VF size [38]; however, visual function largely correlates with fundus appearance. Patients with a normal fundus or tapetal reflex are likely to maintain their VA; one study reported that only 7% of patients with these appearances had reduced VA [36]. VA deteriorated at a rate of 1.4% per year in females with focal/limited pigmentary changes and at a rate of 1.9–2.3% per year in females with diffuse or more widespread changes [36,38]. A high proportion of adult females with XLRP manifest significant full-field electroretinography (ERG) abnormalities due to generalized rod and cone system dysfunction, although the changes are usually much milder compared to male patients. In addition, asymmetry between the eyes might be seen [21]. Further investigation of females with *RPGR* variants may be needed to fully understand the disease expression in females.

We present the specific genetic characteristics and long-term follow-up of Slovenian patients with *RPGR* retinopathy.

## 2. Results

Eight families exhibited RP (nine males with typical RP and six females with various phenotypes), while two families (three male patients) exhibited COD. None of the variants was associated with both phenotypes. The genetic and clinical findings are summarized in Table 1 and Table 2, respectively, and described in detail below. Family pedigrees are shown in Figure 1.

### 2.1. Genetic Findings

The locations of the identified *RPGR* variants in Slovenian families are shown in Figure 1. The eight unrelated families that exhibited RP were found to harbor seven different *RPGR* variants: c.1217dupT p.(Ser407Ilefs*46), c.2236_2237delGA p.(Glu746Argfs*23), c.G1978G>A p.(Glu660*), c.457G>A p.(Ala153Thr), c.1506+1G>T, c.2340_2341delAG p.(Arg780Serfs*54), and c.1245+704_1415-2286del (ex-11del); the last of these was found in two families. The first two variants had been previously reported in the literature and/or submitted to ClinVar (see Table 1), while the other five variants were novel. The two unrelated families that exhibited COD harbored the *RPGR* variant c.3457T>A p.(Ter1153Lysext*38). They were described in detail in a previous publication [39] and are reviewed here for completeness.

### 2.2. Clinical Findings

#### 2.2.1. Retinitis Pigmentosa

The RP cohort consisted of nine affected males (from eight different families) with RP and six females (from four different families) with varying degrees of retinal dystrophy. Due to milder disease in females, their clinical characteristics are described separately.

##### Male RP Patients

All males exhibited nyctalopia. The median age at onset was 6 (range 0–18) years. At their first exam at the median age of 32 years (range 11 months–63 years), their median BCVA was 0.30 logMAR (range 0.15–2.0). All had constricted VF, reduced color vision, and a hyperautofluorescent ring in the macula encircling preserved photoreceptors on optical coherent tomography (OCT) (Figure 2, Figure 3, Figure 4 and Figure 5). Full-field ERG (performed in 4/9 patients) showed reduced rod and cone function in the pattern of rod-cone dystrophy in one patient (F5P10—age 1 year) and complete loss of rod and cone function in three patients (median age 16; range 8–61 years). Microperimetry was performed in one patient (F5P11; Figure 6) who showed complete functional loss of photoreceptors in the fovea. At the last follow-up at the median age of 39 years (range 4–71), the median BCVA was 0.48 (range 0–2.30) logMAR. Color vision had deteriorated in all patients with available data (N = 4) (median follow-up 9 years; range 4–21) from median 9/15 to median 6/15 recognized Ishihara plates. The VF area in patients with available data (3/9) constricted from median 34317 square degrees (range 11767–67738) to 12373 square degrees (range 5246–20752). FAF showed ring constriction in all cases and transition to hyperfluorescent patch in two of these nine cases (Figure 2 and Figure 3). Ring constriction was estimated at the median rate of 0.015 mm^2^ (range 0.014–0.015, N = 4) per year, and median horizontal inner segment ellipsoid (ISe) band loss was estimated at 45 μm (range 34–105) per year (Appendix A). A Kaplan–Meier analysis estimated that 50% of the patients reached legal blindness based on VA at the age of 61 years (95% of patients between the ages of 43 and 79 (Figure 7) and based on VF at the age of 27 years (95% of patients between the ages of 25 and 29 (Figure 8)). They were mostly (7/9) myopic (median refraction error at first visit in the better eye was −3.75 Dsph (range 0 to −11)). However, in one child the refraction error was +0.25 Dsph in the better eye at the first visit (8 years) and +1.0 Dsph in the better eye at the last visit (12 years) (Table 1).

##### Female RP Patients

Females with RPGR-RP-causing variants (N = 6) exhibited various incomplete patterns of disease (Figure 9, Figure 10 and Figure 11). Two patients had a normal/near-normal FAF pattern (F1P3—age 10 years and F2P7—age 35 years), one had a radial pattern (F4P9—age 6 years), two had a focal pattern (F1P2—age 62 years and F6P13—age 66 years), and one had a combination of focal pattern on one eye and male pattern on the other eye (F2P5—age 52 years). One patient (F1P2) with a focal pattern had a phenotype similar to sector RP that exhibited a partial FAF ring (Figure 9). At the first exam at the median age of 40 (range 6–66) years, their median BCVA was 0.0 (range 0.0–0.30) logMAR. Color vision was reduced in 4/5 in whom it was measured. The patient with normal color vision had a normal/near-normal pattern on FAF (Patient F1P3; Figure 9). VF in patients with a normal/near-normal FAF pattern was normal in one and with a few locations of reduced sensitivity in the other. The patients with a focal FAF pattern had relatively large confluent but asymmetrical areas of reduced sensitivity, and patients with radial and male FAF pattern had constricted visual fields. Microperimetry was performed in the patient with sector RP (F1P2; Figure 12) and showed good central fixation and preserved retinal sensitivity in the superior portion of the retina. Follow-up data was available for three female patients (median 4 years of follow-up; range 4–21 years). By the last follow-up at the median age of 45 (range 10–67) years, BCVA had worsened in all (see Figure 13), the median VF area constricted from 747706 square degrees (range 251130–1102152) to 141071 square degrees (range 105834–176308), and color vision worsened in 1/3 patients (F1P2, Figure 9) with available data. FAF showed progression in 2/3 patients (four eyes) with follow-up data (F1P2 and F2P5; see Figure 9 and Figure 10).

Three female patients were followed up with imaging. Among the three eyes that exhibited a complete or partial FAF ring (F1P2 and F2P5), the ring showed constriction. Two eyes progressed to a hyperautofluorescent patch (after a median of 13 years; range 4–21) accompanied by a loss in photoreceptors in the fovea (Figure 9 and Figure 10). The third patient with follow-up images (F4P9; Figure 11), who exhibited a radial pattern, showed no obvious change in FAF after 4 years. Horizontal ISe loss (measured longitudinally in both eyes of three patients with radial, focal, and male FAF patterns) was estimated at an average of 31 μm (27–35 μm) per year (Appendix A). Three out of six female patients had a refraction error (F1P2, F2P5, and F4P9). Two were myopic with a median refraction error at first visit in the better eye −5.0 Dsph (−1.0 to −9) and showed no worsening at the last follow-up (median −5.0 Dsph in the better eye) (see Appendix A). A child with a radial FAF pattern had a refraction error +0.25 Dsph at the first visit (6 years) and −0.5 Dsph at the last visit (10 years) (see Table 1 and Appendix A). Full-field ERG in patients with focal and male FAF pattern showed asymmetrically reduced rod responses and preserved but reduced and delayed cone responses (F1P2 had reduced and delayed DA and LA ERG in both eyes but a normal pattern ERG (PERG) P50 amplitude and multifocal ERG (mfERG) in the right eye, whereas in the left eye she had reduced to undetectable PERG P50 and mfERG. Patient F2P5 had significantly reduced LA, DA ERG, and PERG P50 in her right eye but undetectable LA, DA, and PERG P50 in her left eye; whereas patients with a radial FAF pattern had reduced rod responses (reduced DA ERG) and normal cone responses (normal LA, mfERG, and PERG P50). In the two patients with normal/near normal FAF and normal VF, ERG was not performed. None of the female patients reached legal blindness based on VA or VF by the last exam (ages 10–66) (see Figure 7 and Figure 8).

#### 2.2.2. Cone Dystrophy

The COD cohort consisted of three affected males from two families who harbored the same variant. They were presented in detail in a previous publication [39] and are reviewed here in short for comparison with RP phenotypes. Their median age at onset of visual loss was 25 (range 3–28) years. At their first exam at the median age of 35 (range 16–42) years, their median BCVA was 1.00 logMAR (range 0.20–1.60). All had central VF loss and reduced color vision. FAF showed hyperautofluorescent rings centered at the fovea and encircling photoreceptor loss on OCT (Appendix A). A poorly defined outer segment layer with an unrecognizable interdigitation layer was observed outside the area of atrophy (Appendix A). ERG in all patients showed reduced LA responses, while DA response was normal. FAF showed a hyperautofluorescent ring centered at the fovea that corresponded to the area of photoreceptor loss on OCT. Microperimetry was performed in all COD patients (see Figure 14) and showed patient F10P17 still fixated with the fovea, which indicated the presence of residual photoreceptors in that area, while patients F9P16 and F10P18 shifted the fixation outside the central lesion. At the last follow-up at the median of 42 (range 33–52) years, their median BCVA was 1.30 logMAR (range 1.00–1.50), their median VF loss was 210868 (111371–310364) square degrees, and FAF showed enlargement of the rings’ diameters on average of 80 μm per year (Appendix A). ERG in patients F9P16 and F10P18 with available follow-up data showed significantly reduced to undetectable PERG P50, significantly reduced mfERG, undetectable LA ERG, but normal DA ERG. Their female relatives were unaffected. All male COD patients were myopic. OCT showed horizontal ISe loss of a median 77 μm per year (Appendix A). The Kaplan–Meier analysis predicted that 50% of patients reached legal blindness based on VA at 42 years of age (95% of patients between the age of 24 to 60 (Figure 15)). In families with COD patients, there were two (possible) females affected according to the pedigrees (Figure 1): a sister of patients F10P17 and F10P18 (see Table 1) who was not examined, but family history data stated that she had a high refraction error without any other symptoms; and the mother of patient F9P16 (see Table 1), who had a genetically confirmed *RPGR* variant but with a normal exam that included FAF (Appendix A).

#### 2.2.3. Comparison between Male RP and male COD Patients

We compared the phenotypic characteristics of male RP and male COD patients. Females were excluded from this analysis due to incomplete presentation associated with random X chromosome inactivation and other factors described above (Appendix A).

The age at onset of visual symptoms was significantly lower in RP patients (median 6 vs. 25 years, *p* = 0.035, log-rank (Mantel–Cox) test) (see Figure 16). Figure 13 shows the longitudinal data on VA from both groups. The survival analysis showed that the age when 50% of patients reached legal blindness based on visual acuity (VA ≤ 1.0 logMAR) was significantly different between male RP and male COD patients (62 years and 42 years, respectively; *p* = 0.046, log-rank (Mantel–Cox) test) (see Figure 15). FAF showed hyperautofluorescent rings in both groups that delineated the border between the relatively preserved and affected retina; the difference was that in the RP patients the photoreceptors were preserved inside the ring, while in the COD patients it was preserved outside the ring. The rings of RP and COD patients were located in a similar region: 152–3318 μm from the fovea (Appendix A). The horizontal Ise loss on OCT was estimated to be 45 μm per year in RP cases and 77 μm per year in COD patients (Appendix A). ERG in male RP patients showed a reduced to undetectable LA ERG, DA ERG, and PERG P50; whereas in male COD patients ERG showed a significantly reduced to undetectable mfERG, PERG P50, and LA ERG, while the DA ERG was normal. Wide-field (Optos, San Diego, CA, USA) color and autofluorescence fundus images of male RP and COD patients are shown in Figure 17.

## 3. Discussion

The paper describes genetic and clinical characteristics of Slovenian RPGR patients. The majority of patients carried novel variants and had a long-term follow up.

### 3.1. RPGR Variants Identified in the Slovenian Cohort

Among the eight identified variants in *RPGR*, the majority (88%; 7/8 variants; 9/10 families) were truncating, while only one was missense (12%; 1/8 variants; 1/10 families). Out of the eight variants, 75% (6/8) were private to the Slovenian population, while two were reported previously in other populations (Table 2; Figure 1). Four variants presumably affected all three major isoforms, while variants in exon 15 and ORF15 exon (c.1978G>A p.(Glu660*), c.2236_2237delGA p.(Glu746Argfs*23), c.2340_2341delAG p.(Arg780Serfs*54), and c.3457T>A p.(Ter1153Lysex*38)) presumably affected *RPGR^ORF15^* and *RPGR^1-19^* isoforms.

According to the LOVD RPGR database (http://www.LOVD.nl/RPGR, accessed on 15 November 2022), approximately 600 unique pathogenic variants in *RPGR* have been identified to date in association with retinal dystrophy. Of those in the database, the majority (75%) are truncation variants (frameshift 54%; stop variants 21%), whereas missense variants are in the minority (21%). Even higher percentages of truncating variants (84%) were reported in a systematic analysis that included 585 *RPGR* variants [41] as well as in a study that included 234 RPGR patients (89%; of those 69% (161/234) were frameshift and 20% were nonsense) [42], which was consistent with our findings.

Between 60 and 80% of disease-causing variants in *RPGR* are found in the ORF15 exon [22,42,43]. Exon ORF15 is the longest exon of RPGR and contains 567 amino acids or approximately half (567/1152 amino acid residues) of the total length of the main RPGR isoform. Approximately 20–30% of the pathogenic variants are located in the most difficult-to-sequence central region of ORF15 (c.2470-3230; p. 824–p. 1077) [42].

In our study, 50% (4/8) of the variants were located in exon ORF15 (RP and 1 COD case) and 50% (4/8) in exons 1–14 (all RP cases). This could be the result of skewed distribution in a small cohort that contained only 10 families. However, another possibility is that the current sequencing methods that were employed missed the variants in the difficult-to-sequence region, as there was a group of Slovenian patients with retinal dystrophies in whom genetic analysis failed to detect causative variants.

In a systematic analysis, the rare missense and in-frame variants were found to be enriched in the regulator of the chromosome condensation (RCC1)-like domain [41], a tandem repeat structure that resides in the region between exons 3 and 10 (amino acid residues 54–367) [44]. Accordingly, all diagnostic missense variants in the study by Tuupanen S et al. were also located within that region, and the Slovenian novel missense variant p.Ala153Thr resided in the 5th exon, which further corroborated this observation [42].

The frequency of different RPGR variants varied in different cohorts. For example, the most commonly observed pathogenic variants in the study by Tuupanen S and colleagues, which included 234 RPGR patients of different ethical origins (United States—86%, Canada—9%, Europe—4.5%, Latin America—0.2%, the Middle East—0.4%, and the South Pacific—0.2%), were p.(Glu802Glyfs*12) (20/234 cases, 9%), p.(Glu746Argfs*23) (12/234 cases, 5%), and p.(Glu809Glyfs*25) (10/234 cases, 4%), all of which resulted in RP [42]. In other cohorts; for example, a Chinese cohort described by Yang J and colleagues, the most commonly observed pathogenic variant was p.(Glu746Argfs*23) followed by p.(Glu802Glyfs*32) and p.(Glu1010Glyfs*68), all of which resulted in RP [41]. In an Italian cohort described by Di Iorio V and colleagues, the most frequent variant was a newly described large deletion in ORF15 followed by p.(Glu922Glyfs*156) and p.(Glu802Glyfs*32) [45], all of which resulted in RP as well. With the exception of the above-mentioned p.(Glu746Argfs*23), none of these variants were found in the Slovenian cohort. This reflected the variability in the *RPGR* variants and the regional genetic specificity of the *RPGR* patients. The latter was observed previously for other retinal dystrophy genes in Slovenia; i.e., *USH2A* [46], *BEST1* [47], *ABCA4* [48], and *DRAM2* [49]. Reports on small cohorts of specific ethnicities can add important information to the known landscape of genetic variants in specific genes.

### 3.2. Phenotypes Observed in the Slovenian RPGR Cohort

The eight *RPGR* variants in our cohort that resided between the 153rd and 780th RPGR amino acid residues (Table 1) resulted in RP and were associated with an early onset of nyctalopia and peripheral VF constriction. On the contrary, the most distal variant at the 3′ end—p.(Ter1153Lysext*38)—resulted in an adult-onset COD that was associated with central visual loss (reviewed in [39]). Phenotypically, in up to 95% of cases, the resulting phenotype of pathogenic variants in *RPGR* is RP, which usually arises from variants at the 5′ end, while COD/CORD-causing variants localize at the 3′ end [30,31,50,51]. There is a watershed zone between the 949th and 1047th residues that is associated with either RP or COD/CORD 2020 [21]. A rare phenotype of atrophic macular degeneration has also been observed that was associated with a variant in that region (ORF15+1164G>T; p.(Glu973Ter)) [52].

In the Slovenian RPGR-RP cohort, there were nine males with RP and six females with various degrees of asymmetrical retinal degeneration. The observed early age at onset in males (median 6 years; range 11 months–18 years) was in concordance with previous studies that reported a relatively early onset of RP associated with the *RPGR* gene [18,27] in comparison to other RP-causing genes. The main initial symptoms were nyctalopia and constricted VF loss, which is typical for RP. Central visual loss also occurred relatively early, and 50% of patients reached legal blindness based on VA at the age of 62 (Figure 15). This was in concordance with previous observations that *RPGR* is one of the RP-causing genes associated with an early central visual loss [53].

There have been conflicting reports regarding the association between the location of the variant within the *RPGR* gene and the disease severity. Some studies reported that variants in the RCC1-like domain (RLD) of the N-terminus resulted in a more severe disease compared with variants in ORF15 [15,48,49,50], while others did not find any differences between ORF15 and non-ORF15 disease severity based on structural and functional measures [54,55]. On the contrary, ORF15-RP-causing variants were reported to result in better ERG responses and a more intact VF compared with variants in exons 1–14 [17]; however, the opposite was reported by others [18,56]. In a systematic analysis by Yang J et al. that included 62 unrelated families and a total of 46 likely pathogenic *RPGR* variants, more than 85% of the patients had RP, while 15% were diagnosed with a variety of X-linked retinal diseases that included CORD, COD, high myopia, and macular dystrophy [41]. Male patients showed a significant reduction in BCVA with increased age. Their results showed that patients with exon 1–14 variants had significantly better BCVA than those with ORF15 variants (*p* = 0.005). For females, the BCVA also showed a significant reduction with the duration of the disease, but BCVA in females with exon 1–14 variants was not significantly different compared to those with ORF15 variants [41]. These contradicting reports may be in part due to the differences in methodology; e.g., inclusion criteria with some studies including only RP patients and others including all patients with RPGR retinopathy or a difference in the measured parameters (BCVA, ERG, etc.). The distal variants resulted in CORD/COD, which is a distinct disease that is difficult to directly compare with RP. Establishing a correlation between the variant location and the severity of RP in the Slovenian cohort was not possible due to the small number of patients of different ages. Nevertheless, as expected, there were differences between the COD (most distal variant) and RP patients (proximal variants).

In the COD cohort of two families with three male patients, the median age at onset was 25 (range 3–28) years, which was significantly later than in the male RP patients (median 6 years; range 0–18 years). It has been noted that in RPGR-COD, the symptoms began later in life but progressed relatively early to legal blindness based on the loss in VA [18,31], which was also observed in our cohort (Figure 15). The initial median BCVA in male COD patients (median 35 years) was 1.00 logMAR, which worsened to 1.30 logMAR at the median age of 42 years, and 2/3 (66%) patients reached legal blindness. In comparison, the male patients with RP had an initial median BCVA (median age 32 years) of 0.30 logMAR, which worsened to 0.48 logMAR at the median age of 39 years (Figure 15), and only 3/9 (33%) reached legal blindness based on VA. Patients with RP, however, usually reached legal blindness earlier based on VF compared to VA [27]. A longitudinal VF analysis of male RP patients showed that 50% of the patients reached legal blindness at the age of 27 years (Figure 8).

Despite the later disease onset of RPGR-COD, there is early macular involvement, and the rate of VA is hence faster in patients with COD than patients with RP [18]. The differences between RPGR-RP and RPGR-COD/CORD can also be observed using retinal imaging and ERG. FAF imaging in *RPGR* retinopathy in both phenotypes frequently reveals parafoveal rings with increased FAF that delineate the border between the affected and unaffected retina [18,28,57,58]. The important difference is that in RP, the preserved retina is inside the ring (Figure 2), whereas in COD it is outside of the ring (Appendix A). The size of the rings can be used to follow the disease progression. A hyperautofluorescent ring is common to RP of different genetic backgrounds, and studies have shown progressive constriction of the ring with time [57,59,60] and its transformation into a patch and later atrophy [60]. This was also observed in our RPGR-RP cohort, in which the ring diameter constricted by an average of 54.5 (range 15–127) μm per year and transformed into a patch in 2/9 cases. On the other hand, the rings of hyperautofluorescence in COD/CORD that encompass the area of degenerating retina [18,28,57] may enlarge with time [29,57,61]. This was also observed in our study, in which the ring diameter enlarged by an average of 80 (range 7–147) μm per year. It was interesting to observe that although they corresponded to different patterns of degeneration, the rings of RP and COD patients were located in a similar region (Appendix A). This could be related to the anatomical aspects of the cone/rod density, which does not change linearly throughout the retina. There is a rod-free zone within the central 350 μm, an approximately equal number of rods and cones between 400–500 μm, and a sharp decline in cones with a concomitant increase in rods peripheral to that region [62]. It is possible that the degeneration of the primarily affected cell (either a rod or cone) occurs faster in the region of their highest density and then proceeds at a slower rate. On the other hand, as related to RPGR-COD, because cones are scarce in the periphery, their loss in that region may not be obvious on imaging; e.g., it may not result in hyperautofluorescence and/or ISe loss on OCT. However, the horizontal ISe loss on OCT was in RP cases estimated at 45 μm per year and in COD patients at 77 μm per year, which may also suggest faster structural deterioration in the macula in COD patients (Appendix A).

In patients with RPGR-RP, the full-field ERG is typically severely subnormal with delayed peak times from childhood. Young adults usually have severely abnormal or undetectable DA and abnormal LA responses [21]. There is frequently early macular dysfunction that manifests as a PERG P50 amplitude reduction [21], which is atypical in comparison to other genetic types of RP; e.g., RHO-RP [25,53]. In Slovenian RPGR-RP patients, the ERG showed abnormal rod system responses, while cone system responses and PERGs were delayed or undetectable, which was consistent with RP with early macular involvement. In RPGR-COD/CORD, LA ERGs are typically delayed and/or reduced, and in CORD there is involvement of the DA ERGs. There is early and severe macular involvement that is characterized by a PERG P50 amplitude reduction. In young and mild cases with relatively small rings of increased parafoveal FAF, the PERG P50 might be relatively spared [63]. In our COD patients ERG showed reduced cone system responses, rod system response was normal and PERG P50 was reduced or undetectable.

### 3.3. Females Harboring RPGR Variants

A wide range of retinal phenotypes has been reported in females with pathogenic RP or COD/CORD-causing *RPGR* variants, which probably reflects random X-inactivation and other factors in gene expression. Between 30 and 60% have neither symptoms nor evidence of retinal pigmentary changes, whereas 40–70% are affected to various extents, and approximately 25% manifest RP or COD/CORD. The disease is usually milder compared to male patients [21,36]. In their study that involved females with XLRP for RPGR, RP2, and some who were not genotyped, Comander J and colleagues found that 40% of them showed a baseline abnormality on at least one of the tests (VA, VF, or DA ERG). In all female patients, the average ERG amplitude to 30 Hz flashes was about 50% of normal, and the average exponential rate of amplitude loss over time was half that of XLRP males (3.7% per year vs. 7.4% per year) [36]. Another study by Talib M and colleagues also reported that females with RPGR variants showed significantly reduced ERG amplitudes in various patterns in 42 out of 59 (71% heterozygotes) [38]. In our female cohort, 2/6 had near-normal fundus, whereas 4/6 (67%) had obvious signs of retinopathy. There were no differences in variant distribution or variant type between male and female patients in the literature (including RP and COD/CORD patients) [42]. Interestingly, in the present study, the affected females were only from families with RP-causing variants, while none of the 2 COD-variant females had a disease phenotype (with the exception of myopia). The variant p.(Ter1153Lysext*38) was at the very end and possibly affects females. However, a larger cohort of females with the latter and other COD-causative variants would be needed for further research on this field.

For RP-causing RPGR variants, different patterns of retinal pathology have been described in females, which was reviewed in Introduction. Visual function largely correlates with fundus appearance in females. Patients with a normal fundus or tapetal reflex are likely to maintain their VA; one study reported that only 7% of patients with these appearances had reduced VA [36]. A high proportion of adult females with XLRP manifest significant abnormalities in DA and LA ERG responses in keeping with generalized rod and cone system dysfunction, although this is usually much milder than in male patients. Inter-ocular asymmetry may be seen, and this may prompt investigation of the genetic status in patients examined prior to genetic confirmation [21]. In our cohort, notable asymmetry was present in 2/6 patients (Figure 9 and Figure 10). The other sign of X-inactivation may have been the radial appearance of retinopathy with exchanging streaks of affected and unaffected retina (Figure 11).

There have been conflicting reports for females regarding a correlation between the location of variants within the *RPGR* gene and visual function, which was noted in male patients. One study reported that females with variants in *RPGR^ORF15^* had lower LA 30Hz b-wave ERG amplitudes compared to females with variants in exons 1–14 [36]. Another study doubted that there was no correlation between the sequence variant site and the phenotype when *RPGR* variants were categorized according to RP versus cone-rod involvement [38]. Our affected females (4/6) had a focal or male pattern of RP. FAF showed concentric hypoautofluorescent spots with hyperautofluorescent areas and retinal atrophy, and ERG showed reduced and delayed DA and LA ERG as well as reduced to undetectable PERG P50 and mfERG.

When comparing both eyes, BCVA and FAF showed more interocular symmetry in male than in female patients (Figure 2, Figure 3, Figure 4 and Figure 5 and Figure 9, Figure 10 and Figure 11). Furthermore, the results showed a faster deterioration in the vision field in males compared to females.

### 3.4. Pathogenesis of RPGR-Associated Retinal Dystrophies

The RPGR protein is thought to be involved in ciliary transport [53,64], which is important to myriad proteins that are crucial to photoreceptor structure and function. Mutations in many of other ciliary genes (as well as genes that transcribe the transported proteins) are also associated with RP. This suggests a shared disease pathway in different types of RP; however, the exact mechanism is not yet understood. In all cases, the rods degenerate first, and degeneration of cones is secondary, although in some cases (including RPGR) relatively early [18]. One of the hypotheses to explain secondary cone degeneration proposed the involvement of reactive oxygen species (ROS), which are the small and highly reactive molecules formed naturally as an expected product of oxygen metabolism in the mitochondria [65]. At physiological levels, ROS act as signalling molecules, but in some conditions the levels of ROS can increase and generate oxidative stress in the cell. The retina is one of the most metabolically active and oxygen-consuming tissues of an organism, but at the same time it is very vulnerable to oxidative stress. It contains a large number of mitochondria and is under constant stress due to photochemical reactions [66]. Rods are metabolically active cells with a high level of oxygen consumption. Choroidal vessels are not autoregulated by tissue oxygen levels, and as rods die, the tissue level of oxygen in the retina increases [67]; this is manifested by the narrowing of retinal vessels, which are autoregulated. Some studies suggest rod cell death may increase the oxidative stress in the retina and consequently enhance the oxidative damage and death of cones as well as rods in RP cases [68]. A study by Shen et al. demonstrated in a pig model of RP that after rods degenerated, macromolecules in cones showed evidence of oxidative damage [67]. The most striking increase was seen in markers of lipid peroxidation, which were localized predominantly in cone inner segment, cone axons, and cell bodies [67]. Markers of lipid peroxidation are often the most prominent indicator of oxidative damage involved in pathogenesis of the disease [67]. Additionally, oxidative damage was seen in proteins, carbohydrates, and DNA.

The COD/CORD phenotype associated with RPGR is perhaps even less understood. It is not clear whether oxidative stress may play a similar role in COD/CORD disease pathogenesis as well as in RP. For execution of transport function, the RPGR protein interacts with other ciliary proteins such as RPGR interacting protein 1 (RPGRIP1), centrosomal protein 290 (CEP290), nephrocystin-5 (NPHP5), and nephrocystin-6 (NPHP6) [69], which differ among rods and cones. However, it seems that different variants in *RPGR* can result in different protein-lacking roles in rods and cones [31]. Another study suggested that mislocalization of rhodopsin and cone opsin early in the disease may render rods and cones more prone to the influence of additional pathogenic modifying effects [64].

Furthermore, a study by Donato L and colleagues revealed another possible mechanism of regulation based on greatly modulated post-transcriptional mechanisms such as alternative splicing and RNA modifications [70]. It was suggested that oxidative stress can modify RNA sites that belong mainly to genes involved in the intracellular anatomical structure pathways, especially in the cytoplasm and nucleus, which can therefore lack their function in the cell. These hypotheses led to the conclusion that treatment with antioxidants may be beneficial to patients with RP [70].

### 3.5. Significance and Novelties of the Slovenian RPGR Study

Although several previous studies focused on RPGR-associated retinal disease, the present study provided several novel findings. The most important novel findings were the pathogenic variants that were identified for the first time in a Slovenian cohort and brought to light to the genetic spectrum in this region. Furthermore, the study strengthened the observation that although male COD patients’ disease onset is later than that of male RP patients, they progress earlier to legal blindness. This suggests that RPGR-COD and RPGR-RP have a distinctly different pathogenesis and that RPGR-RP cannot be thought of as ‘‘RPGR-COD with an additional rod involvement’’. Further studies to elucidate the specific functions of RPGR in cones and rods are warranted.

## 4. Materials and Methods

### 4.1. Patients

The study included 10 Slovenian families (11 males and 6 females) examined at the Eye Hospital University Medical Centre, Ljubljana, Slovenia. The study was conducted in agreement with the Declaration of Helsinki. Informed written consent was obtained from the patients.

### 4.2. Genetic and Bioinformatic Analysis

Genetic analysis was performed in probands from each family. Genomic DNA was extracted from blood samples according to the standard procedure. Whole-exome sequencing was performed. Sequencing of the defined clinical target was performed using next-generation sequencing on the isolated DNA sample. Briefly, the fragmentation and enrichment of the isolated DNA sample were performed according to the Illumina Nextera Coding Exome capture protocol with subsequent sequencing on an Illumina NextSeq 550 for 2 × 100 cycles (Illumina, San Diego, CA, USA). After duplicates were removed, the reads were aligned to the UCSC hg19 reference assembly using the BWA algorithm (v0.6.3), and variants were called using the GATK framework (v2.8). Only variants that exceeded a quality score of 30.0 and a depth of 5 were used for downstream analysis. Variant annotation was performed using ANNOVAR and snpEff algorithms with pathogenicity predictions in dbNSFPv2 database. Structural variants were assessed using the CONIFER v0.2.2 algorithm. Variants with a population frequency exceeding 1% in gnomAD, synonymous variants, intronic variants, and variants outside the clinical target were filtered out during the analyses. An in-house pipeline was used for bioinformatic analyses of exome sequencing data in accordance with GATK best practice recommendations [71]. The interpretation of sequence variants was based on ACMG/AMP standards and guidelines [40]. When sequencing the DNA sample, we reached median coverage of 67× and covered over 99.9% of the targeted regions with a minimum 10× depth coverage [72]. ORF15 was sequenced directly. The presence of the mutations in the population were examined in the gnomAD database (gnomad.broadinstitute.org, accessed on 14 November 2022).

### 4.3. Clinical Examination

An accurate family history was recorded, and all patients underwent a complete ophthalmic examination that included BCVA (from patients’ data collected in Snellen and then converted to logMAR), an Ishihara test with 15 plates, slit lamp biomicroscopy, and dilated fundus examination. VF was performed using Goldmann perimetry. II/4 stimuli were used for a standard measurement in all patients. Isopters were determined and measured using ImageJ (available online at imagej.net), and the visual field surface was calculated by the program in square degrees (°)^2^. Legal blindness was confirmed if the BCVA in the better eye was ≥1.0 logMAR or if the VF diameter was <20°. The median and range of both eyes’ visual field measurements were calculated. The retinal fundus photographs were conventional 35° fundus colour photographs (Topcon, Tokyo, Japan). Fundus autofluorescence imaging (FAF) (30° and 55° of the central retina) and optical coherence tomography (OCT) extending 8 mm of the macula was performed with a confocal scanning laser ophthalmoscope (Spectralis; Heidelberg Engineering, Heidelberg, Germany). The horizontal diameters and areas of the hyperautofluorescent rings on FAF were measured semiautomatically by manually using the region-finder module available in the Spectralis software. The outer border of the ring was used for the measurement. The integrity of the photoreceptors was determined by qualitatively assessing the inner segment ellipsoid (ISe) band on the OCT. The ISe was measured using the region-finder module in the Spectralis software. Horizontal ring diameters, areas, and ISe bands were measured in both eyes, and the median and ranges of the averages of both were calculated. Wide-field color and FAF imaging (Optos) were performed in selected patients. Pattern and full-field ERG were performed with an Espio visual electrophysiology testing system (Diagnosis LLC, Littleton, MA, USA). The recording electrode was an HK-loop placed in the fornix of the lower lid [73]. Recordings were made according to the standards of the International Society of Clinical Electrophysiology of Vision (ISCEV) [74,75]. The PERG P50 amplitude was used to analyse the macular function. Rod system function was assessed via dark-adapted 0.01 and 3.0 ERG. Cone system function was tested via light-adapted 30 Hz and 3.0 ERG. Multifocal ERG testing was performed according to ISCEV standards [76] with a RETI scan system (Roland Consult GmbH Wiesbaden, Germany). The stimulus included an array of 61 hexagons. Microperimetry was performed in selected patients using an MP-1 microperimeter (Nidek, Padova, Italy), and retinal sensitivity and fixation were assessed.

### 4.4. Statistical Analysis

Statistical analysis was performed using IBM SPSS Statistics 27 software (IBM Corp. Armonk, NY, USA). Legal blindness was assessed using a Kaplan–Meier survival analysis. The comparison of the age at onset of the visual symptoms and the age when 50% of patients reached legal blindness between the RP and COD groups was performed using a log-rank (Mantel–Cox) test.

## 5. Conclusions

This paper reviewed the characteristics and long-term follow-up of 18 Slovenian *RPGR* patients, thereby expanding the knowledge regarding the disease progression in RPGR retinopathy. Since the disease is X-linked, the majority of females with RPGR variants also were affected, and several of them had severe visual loss, which is important to consider in patient counseling. The majority of identified variants (75%; 6/8) had not been previously reported in other *RPGR* cohorts, which suggested the presence of distinct *RPGR* alleles in the Slovenian population.

## Data Availability

The original data are available upon reasonable request from the corresponding author.

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
