# Peer review of "Genetic Characteristics and Long-Term Follow-Up of Slovenian Patients with RPGR Retinal Dystrophy"

_ijms, 2023, doi:10.3390/ijms24043840_

Round 1

Reviewer 1 Report

Overview and general recommendation:

In the research, 10 Slovenian families (11 males and 6 females) which show retinitis pigmentosa (RP) or cone dystrophy (COD) phenotypes are included. The authors first describe the genetic characteristics. Two known variants and five new variants are involved in RP and COD is associated with p.(Ter1153Lysext*38). The authors also performed long-term follow-up of all the patients. The phenotype of male RP patients and female RP patients are described. They also compare the phenotype of male RP patients and male COD patients. The study show how the disease progress in RPGR patients and provide important data in RPGR-retinopathy research.

I find the paper is organized in a proper way and the results are well described. The authors perform background research carefully. And major methods are well described in the manuscript and properly used in the research. All the results are good enough to support the conclusions. Overall, the research lacks significance and novelty. I suggest the authors should explain more about the innovation in the manuscript.

Author Response

Dear Reviewer,

thank you very much for your comments. For our responses, please see the attachment.

Thank you very much.

Best regards,

Authors

Reviewer 2 Report

Hadalin et al. realized a very interesting review describing the "Genetic characteristics and long-term follow-up of Slovenian patients with RPGR retinal distrophy". I consider the manuscript very interesting but, at the same time, i suggest several revisions needed to improve the reliability and the completeness of the paper:

- The introduction sections should be more detailde and improved. I suggest adding data related to the involvment of oxidative stress in the etiophatogenesis of ocular diseases, especially with vascular components. The recent PMID: 32877751, PMID:30523548 and PMID: 36290689 could represent a substrate able to enforce the role of considered cellular mechanisms.

- "The Statistical Analysis" should be better described(an association test between clinical symptoms and genetic variants? was performed?) and lack a post-hoc test (e.g. Bonferroni).

- Finally, manuscript requires important English revisions and typos correction

Author Response

(The authors gave the same response as above.)

Round 2

Reviewer 2 Report

Hadalin et al. realized a very interesting review describing the "Genetic characteristics and long-term follow-up of Slovenian patients with RPGR-retinal dystrophy". I consider the manuscript very interesting but, at the same time, i suggest revisions needed to improve the reliability and the completeness of the paper:

- The "introduction" sections should be more detailed and improved. I suggest adding data related to the involvement of oxidative stress in the etiopathogenesis of ocular diseases, especially with vascular components. The recent PMID: 32877751, PMID: 30523548 and PMID:36290689 could represent a substrate able to enforce the role of considered cellular mechanisms.

- The "statistical Analysis" should be better described (an association test between clinical symptoms and genetic variants was performed ?) and lack a post-hoc test (e.g.Bonferroni)

- Finally, manuscript requires important English revisions and typos correction

Author Response

Dear Reviewer,

thank you very much again for your comments and suggestions. We checked our paper again carefully. Please see our revisions in the file attached.

Thank you very much.

Best regards,

Authors
